# Mass spectrometry reveals the evolutionary conservation of phycobiliprotein complexes

Jaspreet K. Sound [1], Giorgio Bianchini [2], Thrupthi A. Ashok[1], Cecilia Rad-Menéndez [3,4], David H. Green [3], Patricia Sánchez-Baracaldo [2] & Aneika C. Leney [1] ✉

Cyanobacteria are a highly taxonomically and ecologically diverse group of oxygenic phototrophs that have colonized many different environments on our planet. Despite their differences, almost all cyanobacteria rely on highly efficient light-harvesting protein complexes, termed phycobilisomes, for effective photosynthesis. Phycobilisomes, along with the phycobiliproteins that make them up, have maintained their function throughout evolutionary history while also diversifying to optimize energy capture and transfer in different conditions. Here, we use a combination of evolutionary proteomics, phylogenomics, and structural bioinformatics to probe how phycobiliproteins have maintained their function while adapting to different habitats. Using high-resolution native mass spectrometry, we show that the two most abundant phycobiliprotein complexes, phycocyanin and allophycocyanin, are highly dynamic. Moreover, upon mixing phycobiliproteins from cyanobacterial strains representing diverse environments and evolutionary lineages, heterologous phycobiliprotein complexes rapidly form, comprising building blocks from different cyanobacterial strains. Bioinformatics and structural prediction methods allow us to identify critical residues involved in these interactions. We thus demonstrate that key structural features within the phycobiliprotein components have remained conserved over three billion years of cyanobacterial evolution, ensuring effective photosynthesis across a wide variety of natural environments.

Cyanobacteria are a group of photosynthetic prokaryotes that originated over three billion years ago[1–3], making them one of the most ancient and diverse organisms on Earth. As the only prokaryotes capable of performing oxygenic photosynthesis, cyanobacteria were pivotal in the development of the early Earth's atmosphere[4]. Even to this day they play key roles in important global processes such as the carbon cycle[5] and the nitrogen cycle[6]. The cyanobacterial photosynthetic machinery is extremely efficient[7] and involves multiple components, such as the photosystems, the light-harvesting complexes, and the oxygen-evolving complex. Investigating the structural

characteristics and behavior of these components is a key step towards understanding the evolutionary processes that led to their origin.

Cyanobacteria use large antenna complexes for energy transfer during photosynthesis. These complexes, termed phycobilisomes, comprise hundreds of linear tetrapyrrole (bilin)-containing proteins, known as phycobiliproteins, joined together by linker proteins[8]. The careful arrangement of bilin chromophores throughout the phycobilisome enables light energy between the 450–670 nm region to be absorbed and transferred towards chlorophyll within photosystems I and II with high quantum efficiency. While most cyanobacteria possess

[1]School of Biosciences, University of Birmingham, Birmingham, UK. [2]School of Geographical Sciences, University of Bristol, Bristol, UK. [3]Scottish Association for Marine Science, Argyll, UK. [4]Culture Collection of Algae and Protozoa (CCAP), Scottish Marine Institute, Oban, UK. ✉e-mail: a.leney@bham.ac.uk

phycobilisomes, the overall structure and size of these complexes can vary widely across different strains. For instance, rod bundle/paddle-shaped phycobilisomes are found in thylakoid-less cyanobacteria such as *Gloeobacter violaceus*[9,10], while hemi-discoidal phycobilisomes are the most common[11,12], with some hemi-ellipsoidal phycobilisomes arrangements also observed in red algae[8]. In general, phycobilisomes consist of a central core made from the phycobiliprotein allophyco-cyanin, from which rod-like assemblies protrude, comprising multiple copies of the structurally similar phycobiliprotein phycocyanin. In some cyanobacterial species, additional phycobiliproteins, such as phycoerythrin and phycoerythrocyanin, are also present within the rod-like structures, albeit at lower abundance[13]. Within hemi-discoidal phycobilisomes, there is further sub-division into those with di, tri, or penta-cylindrical cores[14].

While there are many variations to the overall phycobilisome architecture[15] across cyanobacteria, the structures of the individual phycobiliproteins are highly conserved due to their specialized function of absorbing and transferring specific wavelengths of light. Each phycobiliprotein is composed of dimeric building blocks consisting of two polypeptide chains, α and β, which assemble into donut-shaped $(\alpha\beta)_3$ hexameric complexes[16,17] (Supplementary Figs. S1, 2). Within the phycobilisome, these $(\alpha\beta)_3$ complexes stack upon one another; their arrangement assisted by linker proteins, which act to hold the phy-cobiliproteins together and aid light energy transmission[18-20]. In cya-nobacteria, the phycobilisome is dynamic and rapidly assembles and disassembles in response to fluctuating light exposure and nutrient levels[21]. Moreover, the allophycocyanin core can even exchange its α and β subunits for alternative variants, α-B and β-18, respectively, to further alter energy transmission from the phycobilisome to photo-system I/II in response to environmental stimuli[22].

Due to the similarities amongst the phycobiliproteins, it has long been suggested that α and β subunits evolved from a single ancestral protein[23-25]. The individual phycobiliprotein complexes then evolved to ensure highly efficient unidirectional energy flow within the phycobilisome[26], whilst increasing their capacity to absorb longer wavelengths of light. However, questions still remain on how the structural framework of the phycobiliproteins has remained conserved in cyanobacteria over billions of years of diversification.

Native mass spectrometry (MS), the analysis of proteins and protein complexes whilst maintaining their native structure[27-29], has proven a fruitful tool within evolutionary proteomics to monitor hemoglobin[30], heat shock proteins[31], and transcription factors[32], but also to determine the evolutionary pressures that have enabled protein complexes to assemble in an ordered manner[33,34]. By focusing speci-fically on proteins that are expressed, native MS enables the ability to draw conclusions from functionally acting proteins, rather than those that are encoded in the genome but are not actively translated. Within the context of phycobilisomes, native MS has been applied to study concentration-based self-assembly of phycocyanin[35], metal-based fluorescence quenching[36], linker protein modifications[37], and to monitor structural changes within phycobiliprotein complexes[38,39] and their associated proteins[40,41]. In addition, through the use of recent advances in high-resolution MS, phycobiliproteins from a mixture of cyanobacterial strains can be rapidly separated and uniquely identified[42].

Here, we use an evolutionary proteomics approach involving native MS to analyze phycobiliproteins from cyanobacterial strains from a range of different environments and evolutionary lineages. We show that across all the strains, phycobiliproteins are dynamic and can rapidly disassemble and reform novel heterologous complexes, com-posed of subunits coming from entirely different species. By mon-itoring the formation of heterologous phycobiliprotein complexes in real time, and combining these findings with structural predictions using AlphaFold2[43], we reveal how photosynthesis within cyano-bacteria has remained a highly dynamic functional process.

## Results

### Phycobiliprotein dynamics vary between phycocyanin rods and allophycocyanin core

We first investigated the intrinsic dynamics within the phycocyanin rods and the allophycocyanin core of the phycobilisome from cyano-bacterial strains found in a range of environments. Eight different cyanobacterial strains were first selected and their phycobiliprotein complexes extracted and analyzed directly by native MS (Table 1, Supplementary Table S1, Supplementary Figs. S3–9). Two mass spectra from the salt-tolerant strains *Limnospira maxima* CCAP 1475/9 (*L. maxima*) and *Gloeocapsopsis crepidinum* CCAP 1425/1 (*G. crepidinum*) are shown in Fig. 1. Native MS analysis of the phycobiliprotein extract from *L. maxima* shows two dominant charge state distributions between 4500 and 5500 *m/z* corresponding to the functional hex-americ $(\alpha\beta)_3$ phycocyanin and hexameric $(\alpha\beta)_3$ allophycocyanin complexes (Fig. 1a). Despite peaks associated with phycocyanin being overall of higher relative intensity, the hexameric $(\alpha\beta)_3$ complex is in dynamic equilibrium with its dimeric αβ building block (Fig. 1a, b). This is consistent with previous studies that have shown that ionic strength[44-46], protein concentration[35] and metal binding[36] can all influence phycocyanin's oligomeric state. On the other hand, allo-phycocyanin from both *L. maxima* and *G. crepidinum* remains hex-americ $(\alpha\beta)_3$, with less than 5% αβ dimer observed relative to the $(\alpha\beta)_3$ hexamer (Fig. 1b, d). Moreover, regardless of the strain chosen for analysis, when both phycocyanin and allophycocyanin are observed, phycocyanin is found to be consistently more dimeric in comparison with allophycocyanin (Supplementary Fig. S3–9).

### Phycobiliproteins from different environments form hetero-logous complexes

Considering the high level of variation in oligomeric states and the structural similarity of phycobiliproteins across cyanobacterial strains, we next sought to determine whether phycobiliproteins from *L. maxima* could co-form complexes with phycobiliproteins from *G.*

**Table 1 | Taxonomic classification and isolation environment for experimentally analyzed cyanobacterial strains**

| Strain | Species | Order | Environment |
|---|---|---|---|
| CCAP 1475/9 | *Limnospira maxima* | Oscillatoriales | Hypersaline |
| CCAP 1425/1 | *Gloeocapsopsis crepidinum* | Chroococcidiopsidales | Marine, thermotolerant |
| CCAP 1453/12 | *Nostoc muscorum* | Nostocales | Freshwater |
| SAMS 01UC | *Kamptonema sp.* | Oscillatoriales | Freshwater |
| ANT.L61.2 | *Phormidesmis priestleyi* | Leptolyngbyales | Freshwater, psychrophiles |
| CCAP 1475/3 | *Spirulina major* | Spirulinales | Brackish |
| CCAP 1403/21 | *Dolichospermum circinale* | Nostocales | Freshwater |
| CCAP 1437/1 | *Gloeomargarita lithophora* | Gloeomargaritales | Freshwater |
| CCAP 1479/10 | *Synechococcus sp.* | Synechococcales | Freshwater |

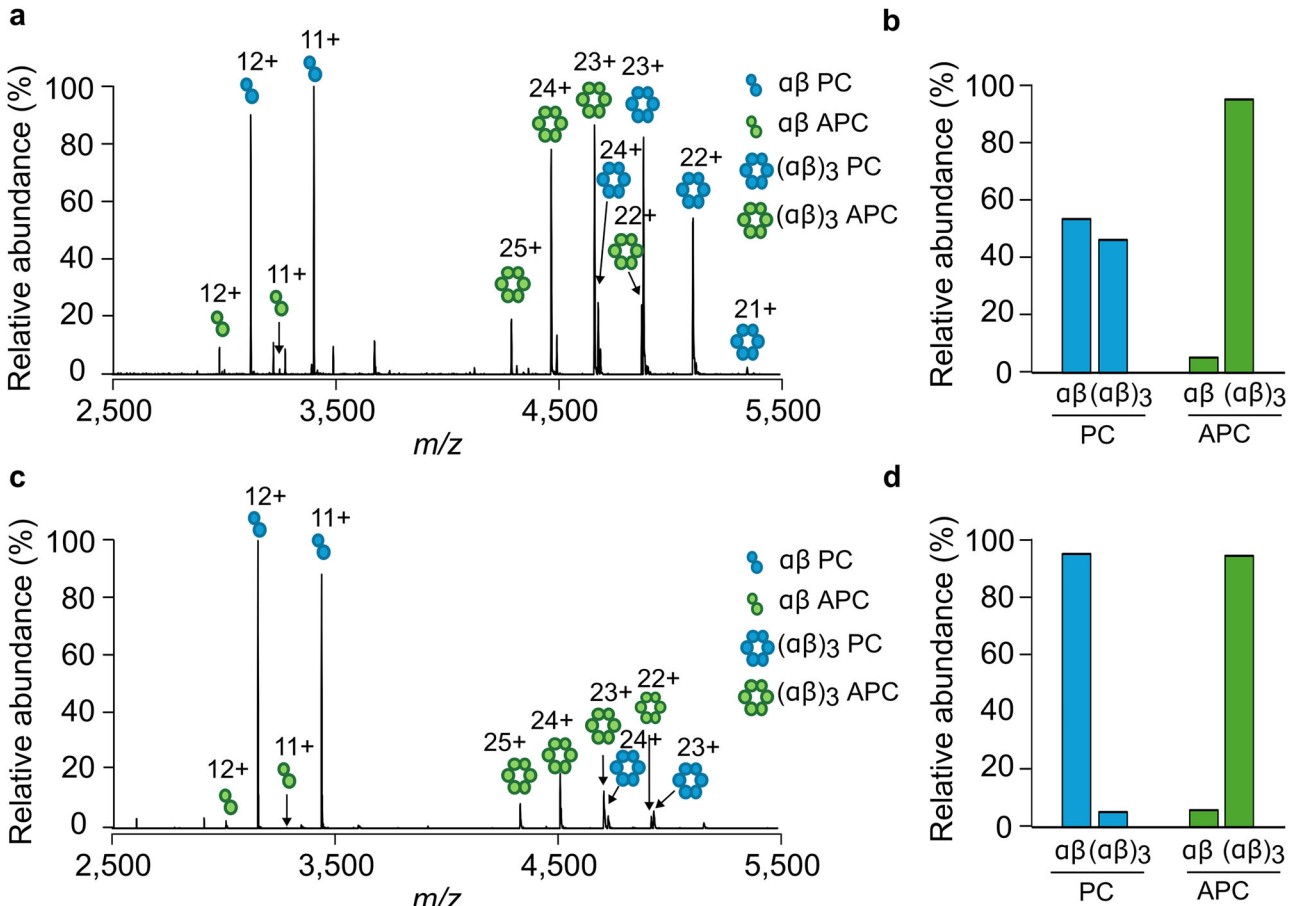

**Fig. 1 | Phycocyanin rods are intrinsically more dynamic than the allophycocyanin core.** Native mass spectrum of *L. maxima* (**a**, **b**) and *G. crepidinum* (**c**, **d**) phycobiliprotein extracts showing the dynamic equilibrium between the dimeric (αβ) and hexameric (αβ)$_3$ phycocyanin (PC, blue) and allophycocyanin (APC, green) phycobiliprotein complexes.

*crepidinum*. To address this, phycobiliprotein extracts from *L. maxima* and *G. crepidinum* were mixed at an equal total protein ratio and the resulting complexes analyzed by native MS (Fig. 2a, Supplementary Tables S2, 3). Due to the high resolving power of the Orbitrap mass analyzer and the subtle differences between the primary sequences of phycobiliproteins from different cyanobacterial strains[42], pure phycocyanin and allophycocyanin from *L. maxima* ((αβ)$_{3xL.maxima}$) and *G. crepidinum* ((αβ)$_{3xG.crepidinum}$) could be clearly resolved and identified. In addition, peaks corresponding to mixed complexes of both phycocyanin and allophycocyanin were observed in the high *m/z* region corresponding to (αβ)$_{2xL.maxima,1xG.crepidinum}$ and (αβ)$_{1xL.maxima,2xG.crepidinum}$.

Next, we chose to determine the extent at which cyanobacterial strains from different environments could form heterologous phycobiliprotein complexes with one another. We mixed phycobiliproteins extracted from the hypersaline strain *L. maxima* and the freshwater strain *Nostoc muscorum* CCAP 1453/12 (*N. muscorum*) (Fig. 2b, Table 1), two fresh water strains with distinct morphologies *Kamptonema sp.* SAMS 01UC (*Kamptonema sp.*) and *N. muscorum* (Fig. 2c, Table 1), and two strains *G. crepidinum* and *Phormidesmis priestleyi* ANT.L61.2 (*P. priestleyi*) that can survive in hot and cold climates, respectively (Fig. 2d, Table 1) and analyzed the resulting complexes by native MS (Supplementary Tables S2, 3). In all cases, when the individual phycobiliproteins were present in the cell lysates from both strains in the mixture, heterologous hexameric (αβ)$_3$ complexes containing αβ dimers from both strains were observed. Thus, this data indicates that if two distinct copies of the phycobiliprotein genes were present and

expressed at the same time within a single strain, mixed phycobiliprotein complexes would form. Interestingly, despite their high structural homology, hexameric (αβ)$_3$ complexes consisting of αβ dimers from both allophycocyanin and phycocyanin were not observed regardless of the strains mixed (Fig. 2, Supplementary Fig. S10), highlighting the specificity in heterologous complex formation. Moreover, the heterogeneity is restrained to the hexameric (αβ)$_3$ complexes, with no mixed complexes evident in the dimer region of the spectrum (Supplementary Fig. S11). We hypothesized that low-abundant linker proteins within our protein extracts could be influencing heterologous complex assembly. Thus, next we took two protein extracts from *L. maxima* and *S. major*, and purified their phycocyanin and allophycocyanin complexes using a combination of ammonium sulfate precipitation and anion exchange chromatography (Supplementary Figs. S12, S13). The highly purified phycocyanin from *L. maxima* formed mixed complexes with phycocyanin from S. major (Supplementary Fig. S14a). Likewise, high-purity allophycocyanin from L. maxima formed mixed complexes with high-purity allophycocyanin from S. major (Supplementary Fig. S14b). Thus, although we cannot rule out that linker proteins may assist in mixed complex self-assembly, mixed complexes can form effectively without their presence.

### Heterologous complex formation occurs in evolutionary distinct strains

To determine the evolutionary relationships between the strains used in Figs. 1 and 2, we performed a phylogenomic analysis (Fig. 3), including the strains within Table 1, as well as a representative sample

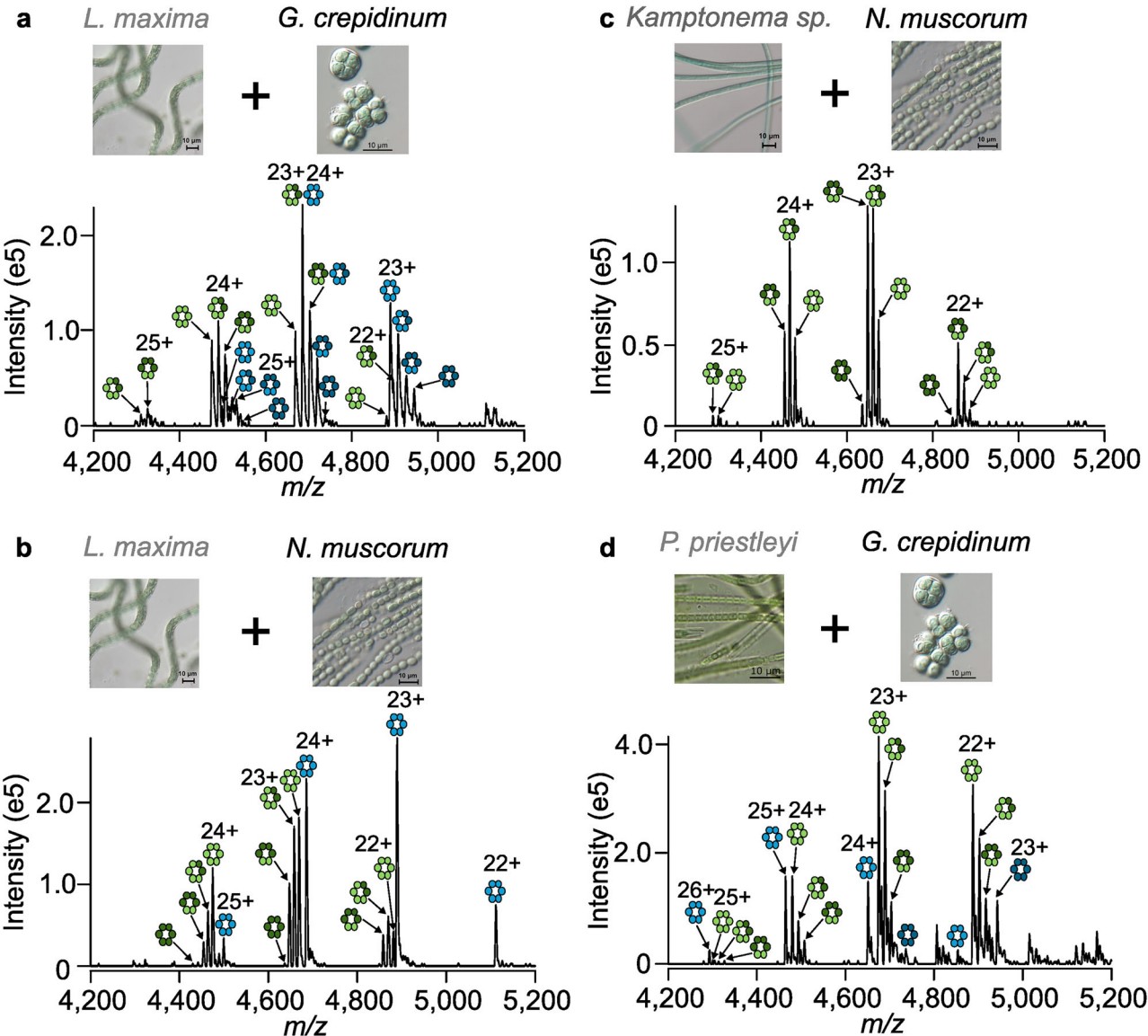

**Fig. 2 | Heterologous phycobiliprotein complexes form containing proteins from different cyanobacterial strains.** Native mass spectra of (αβ)₃ allophyco-cyanin (green) and phycocyanin (blue) complexes formed upon mixing phycobi-liproteins from strains **a** *L. maxima* (light) and *G. crepidinum* (dark); **b** *L. maxima*

(light) and *N. muscorum* (dark); **c** *Kamptonema sp.* (light) and *N. muscorum* (dark); **d** *P. priestleyi* (light) and *G. crepidinum* (dark). The αβ dimeric building blocks are colored (light or dark) according to the strain from which they originate.

of most orders of Cyanobacteria[47]. *L. maxima* and *Kamptonema sp.* belong to the order Oscillatoriales, *N. muscorum* to the Nostocales, *G. crepidinum* to the Chroococcidiopsidales, and *P. priestleyi* to the Leptolyngbyales. Most of our selected strains so far are filamentous (with the exception of *G. crepidinum*, which does, however, form aggregates[48]), and all are members of the Macrocyanobacteria, a monophyletic group that contains strains that generally have a cell diameter >3 μm[49].

To determine whether the phycobiliproteins from evolutionarily distinct strains exchange differently, we performed three additional mixing experiments. Phycobiliproteins from the Nostocales strain *Dolichospermum circinale* CCAP 1403/21 (*D. circinale*) were mixed with the Oscillatoriales strain *L. maxima*, with the Microcyanobacteria strain *Synechococcus sp.* CCAP 1479/10 (*Synechococcus sp.*), and with the early-branching strain *Gloeomargarita lithophora* CCAP 1437/1 (*G. lithophora*), from the order Gloeomargaritales, which represents the group of cyanobacteria most closely related to the chloroplast[50,51]. As

predicted, allophycocyanin αβ dimers from *D. circinale* readily exchanged with those from *L. maxima*, resulting in mixed allophyco-cyanin (αβ)₃ complexes (Fig. 4a–c). Furthermore, despite being from distant evolutionary backgrounds, we also observed heterologous complexes formed from the exchange of αβ dimers between *D. circinale* with *G. lithophora* (Fig. 4d–f) and *Synechococcus sp.* (Fig. S12) for both allophycocyanin and phycocyanin. Across these phycobiliprotein mixing experiments, the degree of heterologous complex formation reflects the difference in allophycocyanin:phycocyanin expression between strains.

**Phycobiliprotein sequence conservation maintains essential function across distinct cyanobacterial lineages**

To investigate structurally how these mixed complexes might form, we used AlphaFold2[43,52] to predict the structure of mixed (αβ)₃ phycobi-liprotein complexes containing αβ dimers from each of the two dis-tantly related strains *D. circinale* (Nostocales) and *G. lithophora*

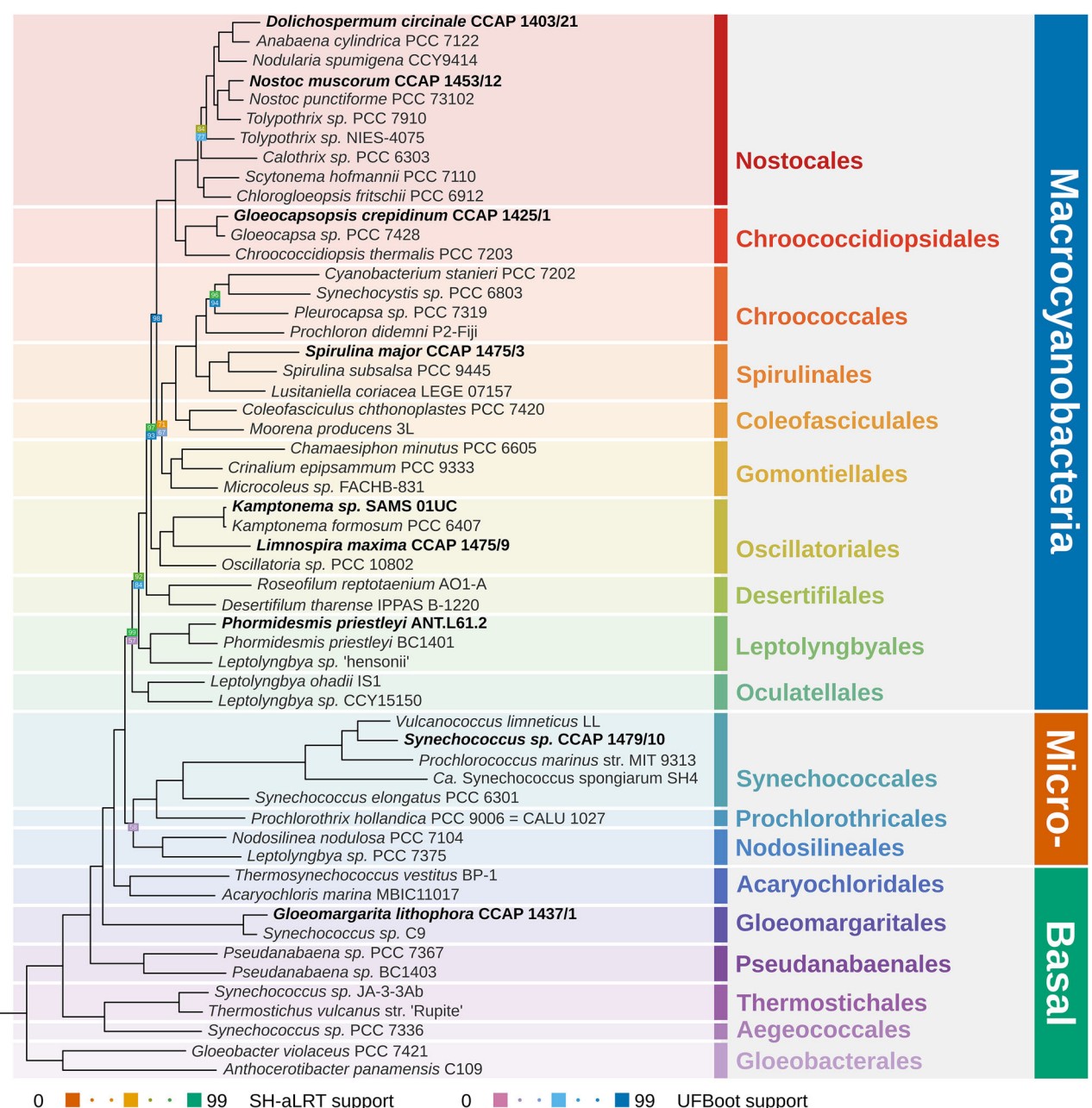

**Fig. 3 | Maximum-likelihood phylogenomic tree of 55 strains of Cyanobacteria.** This is estimated based on 136 protein-coding genes and 16S and 23S ribosomal RNA genes. Strains analyzed in our experiments are highlighted in bold. The SH-aLRT and UFBoot support values are indicated on nodes where they are <100%. The tree was built in IQ-TREE v2.2.5[68] using a 136-gene phylogenomic dataset[1], and visualized using TreeViewer v2.2.0[69].

(Gloeomargaritales), as well as complexes containing only dimers from an individual strain. The structure of allophycocyanin containing two $(\alpha\beta)_{D.circinale}$ dimers and one $(\alpha\beta)_{G.lithophora}$ dimer, a complex observed in our native MS data, is shown in Fig. 5a. To assess the confidence of AlphaFold2 in predicting interactions, we compared the number of contacts observed at the predicted dimer interface within a 3 Å distance for the mixed allophycocyanin structure with the allophycocyanin structure composed entirely from *D. circinale* (Fig. 5a). We also analyzed this using the predicted aligned error (PAE), a measure of the uncertainty in the relative position of two residues, and the predicted local distance difference test (pLDDT), a confidence metric estimating how well the prediction agrees with an experimental structure[43,53]. These measures have been shown to successfully discriminate between interacting and non-interacting proteins[54], and here we used

them to qualitatively compare the binding strength between phycobiliprotein dimers. Consistent with the observation of the mixed allophycocyanin hexamer in vitro, the predicted mixed complex had a low PAE and a high interface pLDDT score, both of which were comparable to those of the *D. circinale* structure alone (Fig. 5b, Supplementary Fig. S16).

Due to the high structural similarity between allophycocyanin and phycocyanin, we next sought to predict why $(\alpha\beta)_3$ complexes containing both phycocyanin and allophycocyanin $\alpha\beta$ dimers were not observed by native MS. Thus, AlphaFold2 was used to predict a structure containing one phycocyanin $\alpha\beta$ dimer from *D. circinale* and two allophycocyanin $\alpha\beta$ dimers from the same strain. The predicted contact interface has a higher PAE, a lower pLDDT, and a smaller number of contacts compared to the $(\alpha\beta)_3$ complexes observed in

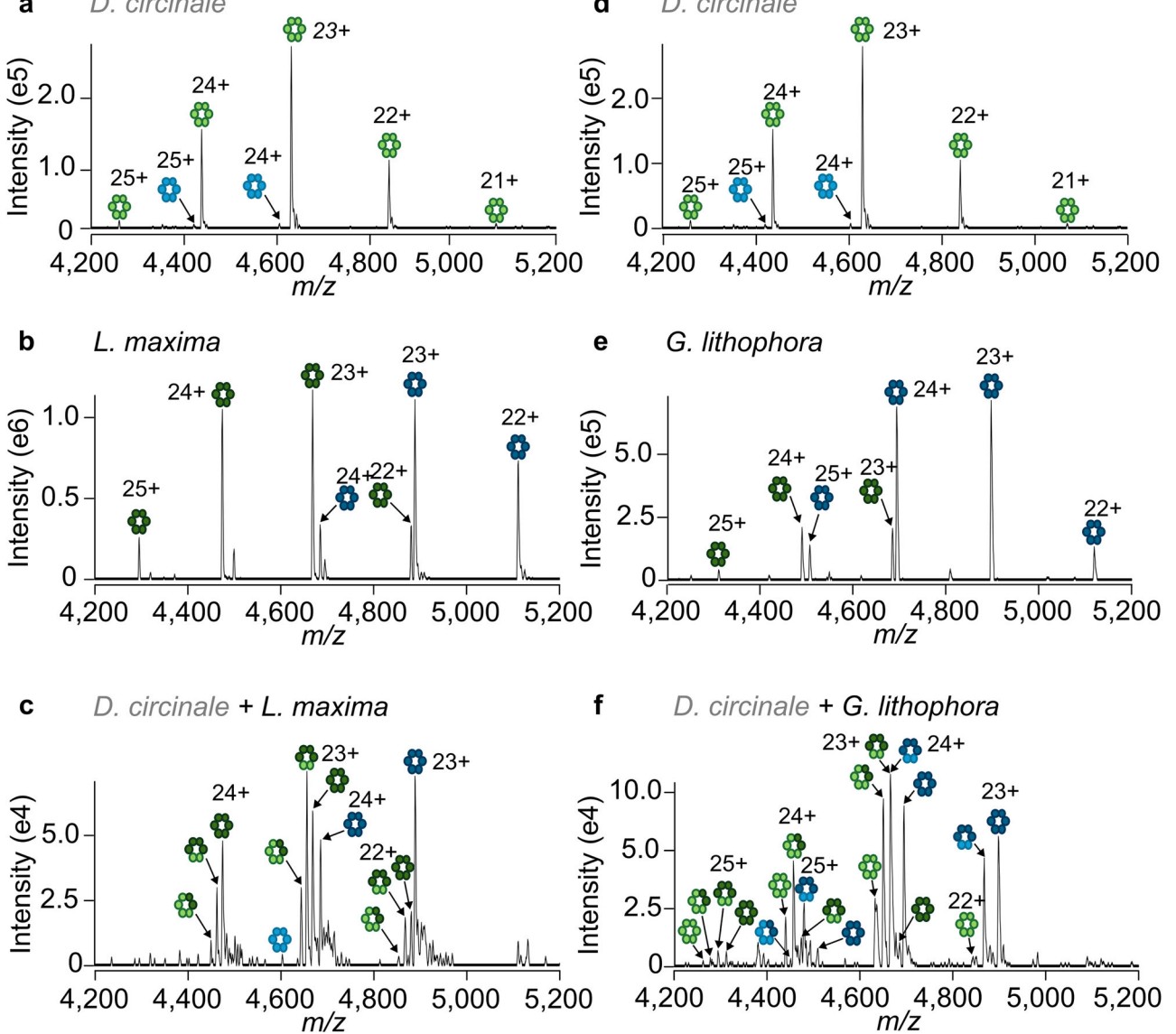

**Fig. 4 | Heterologous phycobiliprotein complexes form containing proteins from distantly related strains.** Native mass spectra of (αβ)₃ allophycocyanin (green) and phycocyanin (blue) complexes from *D. circinale* (**a**, **d**), *L. maxima* (**b**) and *G. lithophora* (**e**) alone, and from *D. circinale* mixed with *L. maxima* (**c**) or *G. lithophora* (**f**). The αβ dimeric building blocks are colored (light or dark) according to the strain from which they originate.

vitro (Fig. 5c, Supplementary Fig. S16). To identify key differences in the residues involved in the contacts between allophycocyanin and phycocyanin dimers, we aligned the sequences of the α and β subunits of allophycocyanin and phycocyanin from 51 strains of cyanobacteria (Supplementary Fig. S23) and inspected conserved residues in the contact interface of the predicted *D. circinale* structures. Helices E and F of the α subunit in both proteins interact with the β subunit's B and E helices and B-E loop. We noted a high degree of conservation of the residues at these critical contact interfaces between αβ dimers across the different cyanobacterial strains, with an average of 97% (99%) and 88% (91%) identity (similarity) in allophycocyanin and phycocyanin, respectively. This supports the fact that heterologous phycobiliprotein complexes comprising subunits from different cyanobacterial strains can be observed by native MS. In addition, we noted that whilst in allophycocyanin a conserved leucine (residue 123 in the alignment, position 119 in the unaligned sequence for *D. circinale*) in helix α-F contacts a conserved tyrosine (residue 83 in the alignment, position 78 in the unaligned sequence for *D. circinale*) in helix β-E (Supplementary

Fig. S24a), phycocyanin has a conserved phenylalanine residue at this position. This different side chain causes the phycocyanin residue to contact with a conserved phenylalanine 62 in helix β-B, rather than with helix β-E as is the case with allophycocyanin (Supplementary Fig. S24b). Furthermore, the allophycocyanin helix α-E has conserved methionine and threonine (rarely serine) residues at positions 81 and 84, which interact with a conserved tyrosine at 63 in the B-E loop of the β subunit. Instead, PC α-E has a lysine at position 84; this amino acid is both has a longer side chain than methionine or threonine and is cationic, and hence the interaction with the B-E loop of the β subunit happens through the isoleucine at position 70. These structural differences highlight why phycocyanin and allophycocyanin do not co-form complexes in vitro. To corroborate this, we predicted the structure of a hypothetical hexamer consisting of two native *D. circinale* allophycocyanin dimers, together with a single *D. circinale* phycocyanin dimer where the regions of the α and β subunits predicted to interact with the other dimers were altered to match those of allophycocyanin. This resulted in a lower interface PAE and higher pLDDT

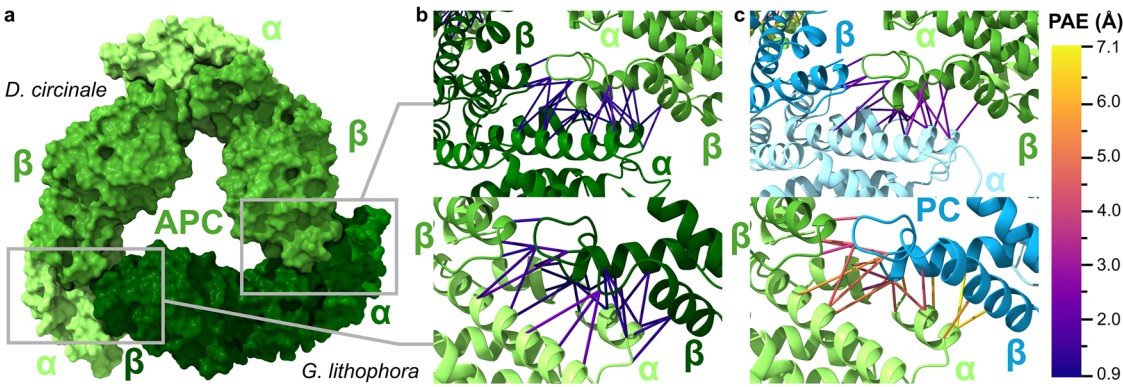

**Fig. 5 | Alphafold models provide rationale behind heterologous complex formation. a** AlphaFold2 model of a heterologous allophycocyanin complex, containing two αβ *D. circinale* dimers (light green) and one αβ *G. lithophora* dimer (dark green). **b** Predicted interfaces between the *G. lithophora* αβ dimer and the two *D. circinale* αβ dimers. **c** Predicted interfaces between a *D. circinale* phycocyanin αβ dimer and two allophycocyanin αβ dimers from the same strain. The predicted alignment error (PAE) for pairs of residues within the dimer interfaces is highlighted.

than the mix of native *D. circinale* allophycocyanin and phycocyanin, as well as the same number of inter-dimer contacts as native allophycocyanin (Supplementary Fig. S16), thus suggesting that this hypothetical complex would be more stable than a mixed hexamer containing native allophycocyanin and phycocyanin. Finally, we sought to investigate the functional advantage of heterologous complex formation. UV-visible absorption spectroscopy data showed different profiles for their phycobiliprotein extracts largely reflecting their abundances by native MS. (Supplementary Fig. S26). Upon mixing phycobiliproteins from two species, the absorbance profile matched the average of the absorbance profile for each individual species. Thus, if a poorly absorbing complex was mixed with a more efficient light energy absorbing complex, its absorbance profile would overall increase, making this beneficial for the organism that originally had an inefficient complex.

## Discussion

Throughout Earth's history, cyanobacteria have evolved and colonized a wide range of habitats, including soil, streams, lakes, oceans, glaciers, deserts, endolithic communities, and hot springs (up to 72 °C)[55]. They also engage in various symbiotic relationships with algae, fungi, plants, and animals. While phycobilisomes play a critical role in the light-harvesting and energy transfer processes during photosynthesis, their evolutionary history remains an open question. Here, we have shown that heterologous allophycocyanin and phycocyanin (αβ)₃ complexes can form consisting of αβ dimers from different cyanobacterial strains. Moreover, this process occurs within minutes and is independent of cyanobacteria strain environment and relatedness. Observations of this nature were only possible due to the high resolving power of the mass spectrometer and its ability to decipher the complex compositions based on the subtle differences in phycobiliprotein masses derived from their amino acid sequences. In fact, suggestions that hybrid complexes can form between unicellular and filamentous cyanobacteria have been previously reported[56,57], however, knowledge on the detailed composition of these heterologous complexes and their quaternary structure was not feasible by the techniques used at that time.

Due to the universal presence of phycocyanin and allophycocyanin within cyanobacteria, it is understood that the two genes encoding the α and β subunits were duplicated before the last common ancestor of crown group cyanobacteria[25], which has been dated to over 3.1 Ga[1–3,58]. Consistent with this hypothesis, our native MS data showed that allophycocyanin αβ dimers do not form heterologous complexes with phycocyanin αβ dimers regardless of the strain they were

extracted from (Supplementary Fig. S10). In addition, the predicted AlphaFold2 structures of heterologous hexamers containing both phycocyanin and allophycocyanin have high PAE and low pLDDT scores compared to their respective homogenous phycobiliprotein complexes (Supplementary Fig. S13). This suggests that ancestral phycobiliproteins were already optimized for function, and capturing energy from multiple wavelengths has always been highly beneficial. We speculate that, at the early stages of evolution, both the phycocyanin and allophycocyanin sets of genes were subject to strong evolutionary pressures. This is because, just after their initial divergence, heterologous complexes could form containing both allophycocyanin and phycocyanin subunits. Thus, structural mutations that initially differentiated phycocyanin from allophycocyanin likely happened on the contact interfaces between different dimers, resulting in the prevention of mixed phycocyanin/allophycocyanin hexamers and thus allowing their divergent evolution.

Overall, our data implies that any phycobiliprotein αβ dimer should be functional in nearly any cyanobacterium in which it is integrated. This is intriguing since phycobiliprotein single gene trees and pan-genome trees are highly concordant, suggesting that phycobiliprotein genes have not undergone significant horizontal gene transfer (HGT). However, excluding these cyanobacterial genes from HGT is unlikely given the significant evidence for HGT in cyanobacteria[59,60] that include core photosystem genes[61]. Moreover, it has recently been shown that the Far-Red Light Photoacclimation (FaRLiP) cluster which contains phycobiliprotein variants can be transferred laterally[62]. Indeed, cyanobacterial cells usually contain multiple copies of their genome per cell[63], meaning that HGT integration events may affect one of many genomes, and the multiplicity of phycobiliprotein genes on the non-integrated genomes dominate the phycobilisomes, maintaining photosystem homeostasis. In support of this, cyanobacterial strains often contain multiple duplicate genes for allophycocyanin[64] and phycocyanin[65] subunits. In addition, there are also allophycocyanin and phycocyanin genes encoding variants involved in phycobilisome assembly and energy transfer (e.g., the α-B subunit encoded by the apcD gene, or the β-18 subunit encoded by apcF) and adaptations to different light intensities or wavelengths (e.g., FARLIP and LOLIP phenotypes[66]). Overall, understanding whether HGT occurs with phycobiliprotein genes and how stringency is enforced remains a subject of further study. Indeed, research into whether heterologous complexes can form in vivo is a critical next step. Regulatory mechanisms may be involved in vivo that either accelerate or prevent heterologous phycobilisome complex assembly. Moreover, by monitoring the assembly/disassembly kinetics of heterologous complexes with linker

proteins present, further insight can be gained into the stability and functional relevance of these complexes in vivo. Nevertheless, if heterologous complexes of this nature can be induced by HGT and are able to form in vivo, this opens the opportunity to biochemically engineer the phycobilisome to change its light absorption characteristics.

## Methods

### Cyanobacterial growth

The strains *L. maxima*, *G. crepidinum* and *S. major* were grown in 50:50 artificial seawater/blue-green (ASW:BG11) culture medium. The strain N. muscorum was grown in BG11$_0$ culture medium. The strains *Kamptonema* sp., *G. lithophora* and *Synechococcus sp.* were grown in BG11. The strain *D. circinale* was grown in Jaworski's Medium culture medium. All CCAP and SAMS strains were kept on a rocker at 20 °C under a 12:12 h light/dark regime. The strains were examined under a Zeiss Axioimager A2 light microscope (Carl Zeiss, Germany). Micrographs were taken with an AxioCam 506 color camera (Carl Zeiss, Germany) using the Zen free software, v2.6.

*P. priestleyi* was supplied by the BCCM/ULC culture collection and grown in BG11 at 18 °C under constant illumination at -10 μmol photons m$^{-2}$ s$^{-1}$. Microphotographs were captured on a Nikon Eclipse Ts2-FL inverted microscope with a 100x objective and a Nikon DS-Fi2 camera.

### Cyanobacterial lysis and phycobiliprotein extraction

Cyanobacterial growth conditions are described in Supplementary methods. Fresh cyanobacterial cell pellets were taken and lysed in an equivalent volume of ultra-pure water using a combination of freeze-thaw (−70 to +25 °C) cycles and sonication. The lysate was then centrifuged at $16,000 \times g$ for 5 min to remove insoluble material. The use of ultra-pure water during cell lysis not only enabled rapid cell lysis, but also ensured a low ionic strength of the solution so that the phycobilisome could dissociate into its phycobiliprotein sub-complexes[44–46]. A visible blue colored supernatant and a strong absorbance reading in 620–650 nm region indicated successful cell lysis. The lysate was then immediately buffer exchanged into 100 mM ammonium acetate pH 6.8 using an Amicon Ultra 0.5 mL centrifugal concentrator with a 30 kDa MWCO (Merck Millipore). This ensured all low molecular weight contaminants were removed. Phycobiliprotein purity was confirmed by native mass spectrometry (Supplementary Figs. S3–9). In all species, the phycobiliproteins peaks in the mass spectrum contribute to ≥65% total ion signal. To further purify phycocyanin and allophycocyanin complexes from L.maxima and S. major to >95% purity, a combination of ammonium sulfate precipitation and anion exchange chromatography was used. For ammonium sulfate precipitation, 25% ammonium sulfate in 5 mM potassium phosphate pH 7.4 was initially added and any unwanted precipitated proteins removed by centrifugation. The phycobiliproteins were then precipitated in 60% ammonium sulfate. Phycocyanin and allophycocyanin were then separated using anion exchange chromatography. The ammonium sulfate precipitated phycobiliproteins were diluted >10-fold into 5 mM potassium phosphate pH 7.4 to remove any residual ammonium sulfate and the proteins loaded onto a self-packed CHT Ceramic Hydroxyapatite column (Type 1, 40 μm) (Bio-Rad) pre-equilibrated in 5 mM potassium phosphate pH 7.4. The absorbance at 280 nm, 620 nm and 650 nm was monitored using ÄKTA pure 25 chromatography system (Cytiva). Phycocyanin interacted weakly with the column and eluted immediately. 50 mM potassium phosphate pH 7.4 was used to elute allophycocyanin. High purity of phycocyanin and allophycocyanin complexes were confirmed by a combination of SDS-PAGE (Supplementary Fig. S12) and native mass spectrometry analysis (Supplementary Fig. S13). The concentration of phycobiliproteins was determined from the absorbance at 280 nm (Jenway 7315 Spectrophotometer) assuming an extinction co-efficient of 1 (mg/ml)$^{-1}$ cm$^{-1}$. All phycobiliproteins extracts were stored at 4 °C in the dark.

### Native mass spectrometry

Phycobiliprotein extracts and purified phycobiliprotein complexes from individual strains were analyzed at a final concentration of 0.2 mg/mL in 100 mM ammonium acetate pH 6.8. To form mixed complexes, phycobiliprotein extracts from different strains were mixed at an equal final concentration of 0.2 mg/mL, incubated for 1 h at 4 °C in the dark, and analyzed immediately by UV-visible absorbance spectroscopy and native MS.

All native MS experiments were performed on an Orbitrap Eclipse Tribrid mass spectrometer equipped with a nano-electrospray ionization source (Thermo Fisher Scientific). For nano-electrospray ionization, borosilicate emitter tips (1.2 mm o.d., 0.68 mm i.d.) were pulled in-house (P-1000 micropipette puller, Sutter Instruments) and gold coated (sputter coater, Agar Scientific). The instrument was operated in positive ion mode and calibrated with FlexMix (Thermo Fisher Scientific). The capillary voltage was set to 1.2 kV, transfer tube at 250 °C, in-source dissociation 0 V and S-lens RF 120 %. Intact protein and high-pressure mode were used throughout. The Orbitrap was used for all mass spectra acquisition, typically with a mass range of 1000–8000 m/z and resolution of 7500 (at 400 m/z). The normalized automatic gain control was set to 100 %, maximum injection time 50–200 ms. All spectra were averaged over 1 min and the data processed using Xcalibur v4.1 (Thermo Fisher Scientific).

Predicted (theoretical) masses of the allophycocyanin and phycocyanin dimers and hexamers for each strain were calculated from the amino acid sequences of the allophycocyanin alpha, alpha-B, and beta chains and phycocyanin alpha and beta chains. The calculated masses were adjusted to include all predicted post-translational modifications (Supplementary Table S1). For allophycocyanin, the mass of the α monomer was modified to include the addition of 1 phycocyanobilin chromophore (+586.7 Da) and the loss of the initiator methionine (−131.2 Da) while the mass of the β monomer was modified to include the addition of 1 phycocyanobilin chromophore and the methylation of Asn71 to N4-methylAsn71 (+14 Da). For phycocyanin, the mass of the a was modified to include the addition of 1 phycocyanobilin chromophore while the mass of the b monomer was modified to include the addition of 2 phycocyanobilins, the methylation of Asn72 to N4-methylAsn72 and between 0 and 1 losses of the initiator methionine. The measured mass error on the experimentally determined molecular weight, and the percentage error between the theoretical and experimentally determined molecular weights were used to verify the presence of the allophycocyanin and phycocyanin complexes (Supplementary Tables S1, S2), whereby percentage errors of <0.1% indicated the presence of the corresponding protein complex.

### Bioinformatics

To perform phylogenomic analysis, a set of 55 annotated cyanobacterial genomes (Supplementary Table S5) was selected, including 9 strains experimentally analyzed in this study and representing 19 cyanobacterial orders[47]. For each genome, we used blastp v2.11.0+ to identify ortholog sequences for a phylogenomic dataset consisting of 145 protein-coding genes; genes for which a single copy ortholog was found in fewer than 30 strains were excluded from the analysis, leaving a total 136 genes. All sequence alignments were performed using MAFFT v7.511[67] with the --localpair --maxiterate 1000 options. The sequences for each gene were aligned and the resulting alignments were manually inspected and trimmed with the online Alignment-Viewer utility, removing positions with more than 85% of gaps and probably mis-aligned regions. The final dataset consisted of 50426 aligned amino acid positions. The best evolutionary model for each gene was selected using IQ-TREE v2.2.5[68,69] with the -m MF --mset LG,Poisson,cpREV,Dayhoff,JTT,WAG,VT,DCMut,PMB,JTTDCMut,-Blosum62,Q.LG,Q.pfam,Q.pfam_gb options, and a maximum-likelihood partitioned analysis was performed in IQ-TREE with the -p option (which allows each partition to have its own evolutionary rate).

Support values for the tree were estimated using ultrafast bootstrap (1000 replicates, with the --bnni option to refine the sampled tree) and the SH-aLRT test (1000 replicates). The analysis was repeated three times and the tree with the maximum likelihood was selected, after verifying that there were no major discrepancies between the trees recovered by each analysis. The tree was visualized using TreeViewer v2.2.0[69].

To investigate phycobiliprotein diversity across the phylum, sequences for the apcA, apcB, cpcA, and cpcB genes (representing, respectively, the α and β subunits of APC and the α and β subunits of PC) were identified using blastp v2.11.0+ with the query sequences listed in Supplementary Table S6. The protein sequences for the five best hits from each genome were retrieved and aligned using MAFFT v7.511 with the --localpair --maxiterate 1000 options, together with the query sequences and with sequences from other known homologs of each gene (Supplementary Table S7). For each gene, a maximum-likelihood phylogenetic tree was built using IQ-TREE v2.2.5 with the --mset Q.pfam_gb option and visualized using TreeViewer v2.2.0, in order to identify orthologs of the gene of interest. The ortholog sequences were re-aligned, and the resulting alignments were manually inspected. Minor improvements were made manually (i.e., aligning the starting methionine residues and correcting the position of loop insertions). A C# script was used to plot the alignment and assess sequence identity and similarity.

Protein structure predictions were carried out using a local install of AlphaFold v2.3.2[43], downloaded with the UniRef30 2023-02 release (Supplementary Figs. S17–22). Structure predictions were run in multimer mode, with template date cutoff 12th December 2023. The protein sequences used for structure predictions are listed in Supplementary Table S8. The predicted structures were visualized and analyzed using UCSF ChimeraX v1.7.1; the amino acid residues on the contact interface between individual monomers were identified using the "alphafold contacts" command. PAE and pLDDT values were extracted from the AlphaFold output files and plotted using a C# script.

### Reporting summary

Further information on research design is available in the Nature Portfolio Reporting Summary linked to this article.

### Data availability

All raw mass spectrometry and UV-vis absorbance spectroscopy data files are freely available via UoB edata archive [https://doi.org/10.25500/edata.bham.00001196]. The *P. priestleyi* ANT.L61.2 genome was deposited in GenBank under the accession JBHLFI000000000 [https://www.ncbi.nlm.nih.gov/datasets/genome/GCF_053471545.1/]. The BioProject number for CCAP/SAMS strains is PRJNA1127564. The CCAP1403/21 genome was deposited under genome accession code JBHYCU000000000 [https://www.ncbi.nlm.nih.gov/datasets/genome/GCA_051861405.1/], CCAP1425/1 under JBHYDD000000000 [https://www.ncbi.nlm.nih.gov/datasets/genome/GCF_051861225.1/], CCAP1437/1 under JBHYDV000000000 [https://www.ncbi.nlm.nih.gov/datasets/genome/GCF_051860525.1/], CCAP1453/12 under JBHYEF000000000 [https://www.ncbi.nlm.nih.gov/datasets/genome/GCA_051860325.1/], CCAP1475/3 under JBIMLG000000000 [https://www.ncbi.nlm.nih.gov/datasets/genome/GCF_051859885.1/], CCAP1475/9 under JBIMLI000000000 [https://www.ncbi.nlm.nih.gov/datasets/genome/GCF_051859755.1/] and SAMS01UC under the genome accession code JBHYGB000000000 [https://www.ncbi.nlm.nih.gov/datasets/genome/GCA_051859225.1/]. Accession numbers for the genomes used in the phylogenomic analysis are provided in Supplementary Table S5; raw, aligned, and trimmed sequence files are accessible from Zenodo (https://doi.org/10.5281/zenodo.17991306). Sequences used for structure prediction are listed in Supplementary Table S8; sequence files, predicted structures, and confidence metrics are accessible from Zenodo (https://doi.org/10.5281/zenodo.17991306).

### Code availability

C# scripts used in the methods and complete sequence alignments are available in GitHub (https://github.com/arklumpus/SoundEtAl) and archived on Zenodo (https://doi.org/10.5281/zenodo.17478265).

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

## Acknowledgements
The mass spectrometry research was supported by the Biotechnology and Biological Sciences Research Council (BBSRC, BB/T015640/1) (A.C.L.). J.K.S. was funded through University of Birmingham funded Midlands Integrative Biosciences Training Partnership (BB/M01116X/1). The Orbitrap Eclipse Tribrid mass spectrometer was funded by the BBSRC (BB/S019456/1) (A.C.L.). We would like to thank Sarah Mansfield-Ford, Jeddidiah Bellamy-Carter, and Pasquale Miglionico for helpful preliminary discussions. Hannah E. Wedgwood assisted with protein purification. We thank the Advanced Mass Spectrometry facility at the University of Birmingham for set-up and maintenance of the mass spectrometers used in this work. Genome sequencing was performed at the Centre for Genomic Research of the University of Liverpool. Genome assembly and phylogenomic analyses were carried out using the computational facilities of the Advanced Computing Research Centre, University of Bristol - http://www.bristol.ac.uk/acrc/. This work was supported by a Royal Society University Research Fellowship to P.S.-B. G.B. was funded by a Royal Society postdoctoral grant given to PSB.

## Author contributions
A.C.L. and J.K.S. designed the mass spectrometry analysis. P.S.B. and G.B. designed the phylogenetic analysis. C.R.M. provided the CCAP cultures and associated images. Cyanobacterial strains were maintained by T.A.A. and J.K.S. J.K.S. collected the mass spectrometry data. J.K.S. and A.C.L. analyzed the mass spectrometry data. G.B. performed the phylogenetic analysis and AlphaFold structural predictions. J.K.S., A.C.L., G.B., P.S.B. and D.H.G. wrote the manuscript with input from all authors.

## Competing interests
The authors declare no competing interests.
