## [Transparent Peer Review file · Nature Communications]

Mass spectrometry reveals the evolutionary conservation of phycobiliprotein complexes

Corresponding Author: Dr Aneika Leney

Version 0:

Reviewer comments:

Reviewer #1

(Remarks to the Author)

Review for Sound et al., "Mass spectrometry reveals the evolutionary conservation of phycobiliprotein complexes."

This work examines the cross-species specificity of phycobiliprotein assembly from (alpha-beta) dimers into hexameric disks, a necessary prerequisite for assembly of the phycobilisomes used for light harvesting in most cyanobacteria. In particular, the work focuses on dimerization and disk formation for the core allophycocyanin proteins (APC; ApcA-ApcB) and for phyocyanin (PC; CpcA-CpcB). Phycobiliproteins were isolated from several cyanobacterial strains (including some whose genomes were sequenced in the course of this study). The formation and composition of dimers and hexameric disks was then monitored by mass spectrometry with or without mixing of proteins from different species to assay whether dimer or disk formation was species-specific. The results show that PC hexamers are notably more dynamic than APC hexamers and that mixed hexameric disks do indeed form. However, mixed alpha/beta dimers and mixed APC/PC hexamers were not detected. The authors interpret the results by modeling both allowed and disallowed hexamers using AlphaFold, using both internal scoring metrics and specific residues. Overall, this is an interesting study that notably demonstrates cross-species disk formation for APC and PC. However, this reviewer does have some questions that should be addressed in a revised version prior to publication.

Major concerns:

1. First and foremost, there is little information presented about these protein preparations, such as an assessment of purity via gel or mass spectrometry. This could be relevant to the results, because it does not seem inconceivable that the presence of small amounts of linkers or bilin lyases could have an effect. For example, linkers from species A might be able to disrupt hexamer formation for disks from species B. Although this does not seem likely, it cannot be ruled out a priori and could lead to a skewed interpretation of the results. Hence, more information about possible contaminating proteins would seem helpful in assessing the conclusions of the work.

2. Similarly, no absorption or fluorescence spectroscopy is presented for these protein preparations. This would again be a useful metric in assessing this work. For example, PC is largely absent from the mass spectra for *Nostoc muscorum* (Fig. S3). Presentation of absorption or fluorescence spectra would distinguish between low levels of PC in this organism under these culture conditions as opposed to poor ionization. Such assays would also be both quicker and cheaper than large-scale transcriptomics or raising antibodies for Western blotting.

Additional points:

Introduction: "all known cyanobacteria contain phycobilisomes" This seems dubious. What about *Prochlorothrix*, *Prochloron*, or *Prochlorococcus*? Some *Acaryochloris* strains also would seem to contradict this statement. *A. marina* MBIC11017 is not representative in this regard, but it also has its phycobiliproteins on a plasmid that also provides additional options for chromophore biosynthesis (Miyake et al. [2020] FEBS J.). The presence of one or more phycobiliproteins need not imply a functional phycobilisome, as shown by cryptophyte algae.

Introduction: Conceptual research into the origins of the phycobilisome precedes the work by Grossman and colleagues; as discussed in that paper, Huber had noted a possible relationship to globins much earlier. That hypothesis has been recently supported by work on phycobiliprotein relatives (BBAGs and photoglobins/photocobilins).

Introduction: The criterion of “successfully expressed and translated” is somewhat artificial for phycobiliproteins; some of them need to be co-expressed as alpha/beta pairs or need to be co-expressed with bilin biosynthesis enzymes and lyases. Moreover, the absence of PC in one of the strains studied here seems at odds with this assertion.

Results: Table 1 could be improved. Which if any of these strains might be able to carry out adaptive responses of the phycobilisome, such as CA/CCA, FaRLiP, or LoLip? Given that some of these strains were newly sequenced for this study, there is not a lot of information about their physiology. Is there any evidence for induction of IsiA under these growth conditions, for example in the Nostoc strain that is lacking PC?

Results: In discussing the strains examined in this study, the authors describe them as “[almost] all filamentous” ... “belong to the macrocyanobacteria” This statement is correct within the context, but it is confusing given the presence of *G. lithophora* in the table and in the phylogeny.

Results: The authors highlight potentially key sequence distinctions that might preclude formation of mixed ApcAB/CpcAB hexamers, notably arguing that Lys has a longer sidechain than Met. However, there are other metrics that might be considered. For example, the Lys sidechain may be cationic, whereas Met would be neutral. This also results in distinct values for hydrophobicity, etc.

Discussion: The authors argue against significant HGT for phycobiliproteins, but this is not supported by any reference. This statement seems worthy of further discussion. It is clear that for example the FaRLiP cluster can be transferred laterally (work from Bryant and Gisriel), and that cluster certainly includes phycobiliproteins.

Overall point for the future: Cyanobacteria in the earliest branches can have duplications in ApcAB or CpcAB. For example, *Anthocerotibacter panamensis* has duplicate ApcAB sequences rather than having ApcD and ApcF. The cryo-EM structure of this atypical PBS confirms that these proteins are present (Jiang et al. [2023] Nat. Commun.). Similarly, members of the Thermostichales (as opposed to Thermosynechococcaceae) apparently have duplicate CpcAB sequences, although in this case there is not experimental confirmation that this is significant. Might it be possible to examine such cases predictively, as done here via AlphaFold? Such an analysis might provide insight into these earlier forms and hence into the process by which most later cyanobacteria adopted a “standard parts list” of phycobiliproteins.

Reviewer #2

(Remarks to the Author)

Phycobiliproteins are the primary light-harvesting proteins in most cyanobacteria and therefore contribute strongly to the molecular basis of how energy enters our biosphere. There are families of phycobiliproteins that serve to harvest different light wavelengths using linear tetrapyrrole chromophores called bilins, and phycobiliprotein complexes often further associate to enlarge the number and wavelength range of photons that can be harvested to drive photochemistry at the photosystems to which they bind.

Despite being in different phycobiliprotein families, allophycocyanin and phycocyanin are surprisingly well conserved both within and across species. This begs the long-standing question of whether phycobiliprotein mixing can occur, which is scantily addressed in the literature. Here, Sound et al. isolate phycobiliproteins from numerous strains of cyanobacteria and use native mass spectrometry to observe that PCs from one species can form PC complexes with those from other species, and the same for APC, but APC never forms complexes with PC. Furthermore, they use a computational approach to determine specific residues that prohibit PC-APC mixing.

This manuscript leverages state-of-the-art techniques to answer fundamental biological questions, and is quite timely considering the recent expansion of phycobilisome morphologies (e.g. paddle-shaped and helical); however, I am reluctant to recommend publication in Nature Communications until the following comments are addressed:

Major:

Based on their native mass spec data, the authors conclude that PC is more “dynamic” than APC. First, I suggest describing this as something other than dynamic or being more specific. To me, “dynamic” implies more flexibility, but I think the authors mean that the oligomeric state is more variable. Although it is possible that PC is more flexible than APC, which would explain why the oligomeric state is more variable for PC, that kind of information is not measured using the techniques herein.

Second, testing various growth conditions (e.g., different light levels) could help to determine whether the oligomeric differences are influenced by external factors or are an intrinsic property of the complexes. At a minimum, this should be discussed.

Third, I was surprised that steady state absorbance spectra were not provided as supplementary information comparing the normal complexes (not mixed from different species) to the chimeric ones (mixed from different species). Do the PC and APC from different species absorb identically? If not, absorbance spectroscopy would nicely complement the native mass spec data. Along the same lines, the authors could have used red-shifted allophycocyanins like allophycocyanin-B to replace the standard allophycocyanin, and the spectroscopic results would have been very compelling. This would be

especially useful in making the argument that by expressing different phycobiliproteins, alterations to the absorbance cross section of an organism could be engineered.

Finally, a main take-home message of this manuscript is exemplified by one of their statements: "Thus, this data indicates that if two distinct copies of the phycobiliprotein genes were present and expressed at the same time within a single strain, mixed phycobiliprotein complexes would form." To me, the mixing of phycobiliproteins in vitro resulting in heterogeneous complexes, which has been performed by the authors, does not necessarily mean that would occur in vivo. It may, but there may also be some regulatory mechanisms involved in assembly that would make in vivo results distinct from in vitro results. I suggest that the authors at least acknowledge this possibility and bring it up as discussion. It seems like in vivo work would also be a logical next step for the author's future endeavors.

Minor:

The authors state "While all known cyanobacteria possess phycobilisomes, the overall structure and size of these complexes can vary widely across different strains." I suggest adding the word "nearly" before "all". To my knowledge, there are some *Acaryochloris marina* spp. that lack phycobiliproteins entirely. There are probably others, too.

The authors state "For instance, rod bundle/paddle-shaped phycobilisomes are found in thylakoid-less cyanobacteria such as *Gloeobacter violaceus*(9, 10), while hemi-discoidal phycobilisomes are the most common(11, 12), with some hemi-ellipsoidal phycobilisomes arrangements also observed." Since the focus of the paper is on cyanobacteria, I think it is worth mentioning that hemi-ellipsoidal phycobilisomes (and also block-shaped) are only found in red algae, not cyanobacteria. The authors may want to cite ref. 8 again here which I think covers the most recently known morphologies.

I suggest rearranging the last three paragraphs of the introduction. Halfway through the third to last paragraph, the phrase "here, we..." is used, and then again at the beginning of the last paragraph. Things could be rearranged to have a smoother transition.

In section "Phycobiliprotein dynamics vary between phycocyanin rods and allophycocyanin core", the authors state "Native MS analysis of the phycobiliprotein extract from *L. maxima* shows two dominant charge state distributions between 4500-5500 m/z corresponding to the functional hexameric ($\alpha\beta$)₃ phycocyanin and allophycocyanin complexes (Figure 1a)." I suggest having "hexameric ($\alpha\beta$)₃" before both phycocyanin and allophycocyanin.

In section "Phycobiliprotein dynamics vary between phycocyanin rods and allophycocyanin core", the authors state "Despite peaks associated with phycocyanin being more abundant overall..." Can the authors clarify this? Are they referring to the number of peaks labeled in, for example, Fig. 4A? If so, it looks like the same number of peaks are labeled for PC and APC. There are more PC peaks than APC in Fig. 4C, but there are less PC peaks than APC in, for example, Fig. S4A.

In section "Phycobiliprotein dynamics vary between phycocyanin rods and allophycocyanin core", the authors state "Despite peaks associated with phycocyanin being more abundant overall, the hexameric ($\alpha\beta$)₃ complex is highly dynamic and observed in equilibrium with its dimeric $\alpha\beta$ building block (Figure 1a,b)." I think the authors are specifically referring to Fig. 1B where the blue bars are nearly equal; however, although this is the case for *L. maxima*, it does not appear to be the case for any of the other species. Maybe my problem is with the author's use of the word equilibrium. Perhaps it would be more appropriate to describe that the ratio of ab dimers being alone or in trimeric complexes is highly variable among species. The APCs being much more commonly in the trimeric form makes sense, so I think that part is fine.

To my understanding, pI-DDT is not a "measure of the overall uncertainty in the position of each residue" as described by the authors, which makes it seem like B-factor in structural biology. Instead, pI-DDT is a confidence metric estimating how well the prediction would agree with an experimental structure. I suggest the authors re-word their description of pI-DDT.

Aside from mentioning them in the introduction, the authors do not mention linker proteins. Some APC subunits have the phycocyanobilin-binding domain but also REP domains that serve as linkers. The authors did not test the ability for these to form chimeric complexes with subunits from other species, but this is something that they may want to mention in the discussion as a future direction.

Similarly, the authors may want to mention that assembly/disassembly kinetic experiments could provide insight into the stability and functional relevance of these mixed complexes.

Reviewer #3

(Remarks to the Author)

Version 1:

Reviewer comments:

Reviewer #1

(Remarks to the Author)

Overall, the revised version is now suitable for publication, and I thank the authors for the extensive work that went into this revision. I do have one concern, pertaining again to the data for *N. muscorum*. In Fig. S19E-F, the spectroscopic data for this strain show a clear peak at ca. 560 nm, which does not match a "standard" or "typical" PC or APC. Instead, the peak in this region would be either PE or PEC. Does this peak arise from an atypical chromophore composition in PC for this strain, or are there additional phycobiliproteins expressed in this strain under these growth conditions?

Therefore, I recommend publication after minor revision.

Reviewer #2

(Remarks to the Author)

The authors have sufficiently addressed our comments and we are now happy to recommend this manuscript for publication.

Reviewer #3

(Remarks to the Author)

Point-by-point response to the reviewers' comments

Reviewer comments are in blue

Author comments are in black

REVIEWER COMMENTS

Reviewer #1 (Remarks to the Author):

Review for Sound et al., "Mass spectrometry reveals the evolutionary conservation of phycobiliprotein complexes."

This work examines the cross-species specificity of phycobiliprotein assembly from (alpha-beta) dimers into hexameric disks, a necessary prerequisite for assembly of the phycobilisomes used for light harvesting in most cyanobacteria. In particular, the work focuses on dimerization and disk formation for the core allophycocyanin proteins (APC; ApcA-ApcB) and for phyocyanin (PC; CpcA-CpcB). Phycobiliproteins were isolated from several cyanobacterial strains (including some whose genomes were sequenced in the course of this study). The formation and composition of dimers and hexameric disks was then monitored by mass spectrometry with or without mixing of proteins from different species to assay whether dimer or disk formation was species-specific. The results show that PC hexamers are notably more dynamic than APC hexamers and that mixed hexameric disks do indeed form. However, mixed alpha/beta dimers and mixed APC/PC hexamers were not detected. The authors interpret the results by modeling both allowed and disallowed hexamers using AlphaFold, using both internal scoring metrics and specific residues. Overall, this is an interesting study that notably demonstrates cross-species disk formation for APC and PC. However, this reviewer does have some questions that should be addressed in a revised version prior to publication.

Major concerns:

1. First and foremost, there is little information presented about these protein preparations, such as an assessment of purity via gel or mass spectrometry. This could be relevant to the results, because it does not seem inconceivable that the presence of small amounts of linkers or bilin lyases could have an effect. For example, linkers from species A might be able to disrupt hexamer formation for disks from species B. Although this does not seem likely, it cannot be ruled out a priori and could lead to a skewed interpretation of the results. Hence, more information about possible contaminating proteins would seem helpful in assessing the conclusions of the work.

The sample preparation for phycobiliprotein extraction is minimal. The cells are lysed in ultra-pure water. This ensures rapid cell lysis, whilst simultaneously decreasing the ionic strength to break the phycobilisome into its phycobiliprotein sub-complexes. The sample is then centrifuged to remove insoluble material, and then buffer-exchanged into 100 mM ammonium acetate pH 6.8. This process involves a 30kDa molecular weight cut-off which acts as an additional level of purification, removing all metabolites, DNA/RNA and most other proteins that are present within cyanobacteria. We confirm the samples are of high purity based off their strong absorbance in the 620-650 nm region corresponding to phyocyanin and allophycocyanin. We also show all the mass spectrometry data of the protein extracts in the SI (Figures S3-9). In all cases, the phycobiliproteins are the most abundant in the spectrum, demonstrating their high purity (>65% for 2500-5000m/z range) considering no

purification was performed. We have now expanded the methods section in the main text to make the extraction and information on phycobiliprotein purity clearer.

We agree that small amounts of linkers or bilin lyases might be present. Indeed, we do see the presence of a linker protein in complex with allophycocyanin in the *P. priestleyi* strain (Figure S5). However, we believe this mixed hexamer formation occurs regardless of whether linker proteins are present or not. To demonstrate this, we have now purified phycocyanin and allophycocyanin from two different strains, *L. maxima* and *S. major*. We used ammonium sulphate precipitation combined with anion exchange chromatography to ensure the highest level of complex purity. SDS PAGE (Figure S12) and mass spectrometry (Figure S13) verified > 95 % phycobiliprotein purity. The newly purified phycocyanin from *L. maxima* formed mixed complexes with phycocyanin from *S. major* (Figure S14a). Likewise, high purity allophycocyanin from *L. maxima* formed mixed complexes with high purity allophycocyanin from *S. major* (Figure S14b). Thus, although we cannot rule out that linker proteins may assist in mixed complex self-assembly, mixed complexes can form effectively without their presence. The main text has been re-worded to include these additional experiments, and the additional data added to the SI.

2. Similarly, no absorption or fluorescence spectroscopy is presented for these protein preparations. This would again be a useful metric in assessing this work. For example, PC is largely absent from the mass spectra for *Nostoc muscorum* (Fig. S3). Presentation of absorption or fluorescence spectra would distinguish between low levels of PC in this organism under these culture conditions as opposed to poor ionization. Such assays would also be both quicker and cheaper than large-scale transcriptomics or raising antibodies for Western blotting.

We have now included UV-vis absorbance spectroscopy data (Figure S20) on all the protein extracts, individual and mixed (requested by reviewer 2), that were analysed by native mass spectrometry in the manuscript. The absorbance spectroscopy traces demonstrate the relative purity of the phycobiliproteins extracted from each of the samples. The absorbance spectroscopy data largely correlates with the relative abundance of phycocyanin and allophycocyanin as determined by native mass spectrometry, however, there are some instances whereby ionisation differences could be coming into play (e.g. for the *Nostoc muscorum* extract). We hope that the additional UV-vis absorbance spectroscopy add clarity to the data presented.

Additional points:

Introduction: “all known cyanobacteria contain phycobilisomes” This seems dubious. What about *Prochlorothrix*, *Prochloron*, or *Prochlorococcus*? Some *Acaryochloris* strains also would seem to contradict this statement. *A. marina* MBIC11017 is not representative in this regard, but it also has its phycobiliproteins on a plasmid that also provides additional options for chromophore biosynthesis (Miyake et al. [2020] FEBS J.). The presence of one or more phycobiliproteins need not imply a functional phycobilisome, as shown by cryptophyte algae.

We agree and should have made this clearer. We have now corrected this sentence to the following:

“While most cyanobacteria possess phycobilisomes, the overall structure and size of these complexes can vary widely across different strains.”

Introduction: Conceptual research into the origins of the phycobilisome precedes the work by Grossman and colleagues; as discussed in that paper, Huber had noted a possible relationship to globins much earlier. That hypothesis has been recently supported by work on phycobiliprotein relatives (BBAGs and photoglobins/photocobilins).

We apologise for this oversight. We have now included reference to this early work within the main text.

Introduction: The criterion of “successfully expressed and translated” is somewhat artificial for phycobiliproteins; some of them need to be co-expressed as alpha/beta pairs or need to be co-expressed with bilin biosynthesis enzymes and lyases. Moreover, the absence of PC in one of the strains studied here seems at odds with this assertion.

Thank you for pointing this out, we have now re-worded this sentence within the introduction:

“By focusing specifically on proteins that are expressed, native MS enables the ability to draw conclusions from functionally acting proteins, rather than those that are encoded in the genome but are not actively translated.”

Results: Table 1 could be improved. Which if any of these strains might be able to carry out adaptive responses of the phycobilisome, such as CA/CCA, FaRLiP, or LoLip? Given that some of these strains were newly sequenced for this study, there is not a lot of information about their physiology. Is there any evidence for induction of IsiA under these growth conditions, for example in the Nostoc strain that is lacking PC?

We understand this information may be interesting for this reviewer, however, we are not focussing on adaptive responses such as CA/CCA, FaRLiP, or LoLip in the main text. We have used standard growth conditions for these experiments. Thus, we do not feel this information is relevant to the Table. We have added a few sentences into the discussion referencing FaRLiP and LoLip highlighting how that this might be an interesting area to expand on in the future (see response to other comments).

Results: In discussing the strains examined in this study, the authors describe them as “[almost] all filamentous” ... “belong to the macrocyanobacteria” This statement is correct within the context, but it is confusing given the presence of *G. lithophora* in the table and in the phylogeny.

We thank the reviewers for pointing this out. We have now made an additional reference to Table 1 and reworded part of the following sentence which we believe will help make this clearer.

“So far, most of our selected strains are filamentous (with the exception of *G. crepidinum*, which does however form aggregates (46)) and all are members of the Macrocyano bacteria, a monophyletic group that contains strains that generally have a cell diameter > 3µm (47).”

Results: The authors highlight potentially key sequence distinctions that might preclude formation of mixed ApcAB/CpcAB hexamers, notably arguing that Lys has a longer sidechain

than Met. However, there are other metrics that might be considered. For example, the Lys sidechain may be cationic, whereas Met would be neutral. This also results in distinct values for hydrophobicity, etc.

We thank the reviewer for this comment, and agree that side chain size may not be the only factor here. We have changed the following sentence accordingly:

“Instead, PC α -E has a lysine at position 84; this amino acid has a longer side chain than both methionine or threonine and is cationic, and hence the interaction with the B-E loop of the β subunit happens through the isoleucine at position 70.”

Discussion: The authors argue against significant HGT for phycobiliproteins, but this is not supported by any reference. This statement seems worthy of further discussion. It is clear that for example the FaRLiP cluster can be transferred laterally (work from Bryant and Gisriel), and that cluster certainly includes phycobiliproteins.

Since the phycobiliprotein single gene trees and pan-genome trees are highly concordant, one would expect that significant horizontal gene transfer (HGT) events have not occurred. As stated in the text, we do think that HGT could have occurred. As such, we originally included the phrase “excluding these cyanobacterial genes from HGT is unlikely”. We have now expanded upon this to include reference to specific papers by Bryant and Gisriel.

“Moreover, it has recently been shown that the Far-Red Light Photoacclimation (FaRLiP) cluster which contains phycobiliprotein variants have been laterally transferred (62).”

Overall point for the future: Cyanobacteria in the earliest branches can have duplications in ApcAB or CpcAB. For example, *Anthocerotibacter panamensis* has duplicate ApcAB sequences rather than having ApcD and ApcF. The cryo-EM structure of this atypical PBS confirms that these proteins are present (Jiang et al. [2023] Nat. Commun.). Similarly, members of the Thermostichales (as opposed to Thermosynechococcaceae) apparently have duplicate CpcAB sequences, although in this case there is not experimental confirmation that this is significant. Might it be possible to examine such cases predictively, as done here via AlphaFold? Such an analysis might provide insight into these earlier forms and hence into the process by which most later cyanobacteria adopted a “standard parts list” of phycobiliproteins.

We have noticed in many cyanobacterial strains that duplicate CpcAB sequences exist within the genomic data. In native mass spectrometry, we see these variations if they are expressed. However, in the strains monitored within this manuscript, as the reviewer correctly states we do not detect any duplicate protein sequences indicating that if present, they are not simultaneously expressed. Hence, for simplicity (as further work is needed as to the reasons behind this) we chose to focus the AlphaFold models and discussion only on what we observed in the native mass spectra. We agree this is interesting though and have now expanded this section within the main text to include this discussion point.

“In support of this, cyanobacterial strains often contain duplicate genes for allophycocyanin (64) and phycocyanin (65) subunits. In addition, there are also allophycocyanin and phycocyanin genes encoding variants involved in phycobilisome assembly and energy

transfer (e.g., the α -B subunit encoded by the *apcD* gene, or the β -18 subunit encoded by *apcF*) and adaptations to different light intensities or wavelengths (e.g., FaRLiP and LoLiP phenotypes (66)).”

Reviewer #2 (Remarks to the Author):

Phycobiliproteins are the primary light-harvesting proteins in most cyanobacteria and therefore contribute strongly to the molecular basis of how energy enters our biosphere. There are families of phycobiliproteins that serve to harvest different light wavelengths using linear tetrapyrrole chromophores called bilins, and phycobiliprotein complexes often further associate to enlarge the number and wavelength range of photons that can be harvested to drive photochemistry at the photosystems to which they bind.

Despite being in different phycobiliprotein families, allophycocyanin and phycocyanin are surprisingly well conserved both within and across species. This begs the long-standing question of whether phycobiliprotein mixing can occur, which is scantily addressed in the literature. Here, Sound et al. isolate phycobiliproteins from numerous strains of cyanobacteria and use native mass spectrometry to observe that PCs from one species can form PC complexes with those from other species, and the same for APC, but APC never forms complexes with PC. Furthermore, they use a computational approach to determine specific residues that prohibit PC-APC mixing.

This manuscript leverages state-of-the-art techniques to answer fundamental biological questions, and is quite timely considering the recent expansion of phycobilisome morphologies (e.g. paddle-shaped and helical); however, I am reluctant to recommend publication in Nature Communications until the following comments are addressed:

We thank for the reviewer for noting the timeliness of our study, and how by monitoring these complexes by mass spectrometry we have made progress in answering a long-standing question of whether phycobiliprotein mixing can occur.

Major:

Based on their native mass spec data, the authors conclude that PC is more “dynamic” than APC. First, I suggest describing this as something other than dynamic or being more specific. To me, “dynamic” implies more flexibility, but I think the authors mean that the oligomeric state is more variable. Although it is possible that PC is more flexible than APC, which would explain why the oligomeric state is more variable for PC, that kind of information is not measured using the techniques herein.

We apologise that this was not clear and thank the reviewer for pointing this out. We were using the word dynamic to explain the equilibrium between the dimeric and hexameric states of phycocyanin compared with allophycocyanin. We have now corrected this to make this clearer.

“Despite peaks associated with phycocyanin being overall of higher relative intensity, the hexameric $(\alpha\beta)_3$ complex is in dynamic equilibrium with its dimeric $\alpha\beta$ building block (Figure 1a,b).”

We have also corrected corresponding sentences in later stages of the text.

From “Considering the high level of intrinsic dynamics and the structural similarity of phycobiliproteins across cyanobacterial strains” corrected to “Considering the high level of variation in oligomeric states and the structural similarity of phycobiliproteins across cyanobacterial strains”

Second, testing various growth conditions (e.g., different light levels) could help to determine whether the oligomeric differences are influenced by external factors or are an intrinsic property of the complexes. At a minimum, this should be discussed.

The oligomeric state of phycocyanin is an important factor in the assembly of phycocyanin into the phycobilisome and is influenced by both external factors and the intrinsic property of the complex. Phycocyanin forms stacked hexamers within the rods of the phycobilisome. Several linker proteins act to stabilise the rod arrangement and ensure unidirectional light energy transfer across the phycobilisome to photosystem I/II. *In vivo*, it has been shown that the phycobilisome adapts to light levels by changing its structure, as nicely reviewed recently in Liu et al. 2024 and elsewhere. *In vitro* studies have also shown that ionic strength (Anderson et al. 1983, Zilinskas et al. 1981, Gantt et al. 1976), protein concentration (Eisenberg et al. 2017) and metal binding (Bellamy-Carter et al. 2022) influences oligomeric state. We have now added a sentence in the results section of the main text to reflect these previous studies.

“This is consistent with previous studies that have shown that ionic strength, protein concentration, and metal binding can all influence phycocyanin’s oligomeric state.”

Third, I was surprised that steady state absorbance spectra were not provided as supplementary information comparing the normal complexes (not mixed from different species) to the chimeric ones (mixed from different species). Do the PC and APC from different species absorb identically? If not, absorbance spectroscopy would nicely complement the native mass spec data. Along the same lines, the authors could have used red-shifted allophycocyanins like allophycocyanin-B to replace the standard allophycocyanin, and the spectroscopic results would have been very compelling. This would be especially useful in making the argument that by expressing different phycobiliproteins, alterations to the absorbance cross section of an organism could be engineered.

As suggested, we have now included absorbance spectra within the supplementary information (Figure S20) and made reference to this data in the main text. This includes absorbance spectra for all phycobiliprotein extracts from each species, in addition to their spectra when two species were mixed. As the reviewer correctly stated, there are differences in absorption profiles for the phycobiliprotein extracts from different species reflecting their abundance within the cells. We see that the absorption spectra of mixed protein extracts which contain heterogeneous phycobiliprotein complexes are an average of the absorption spectra of the protein extracts from individual strains. Thus, if a poorly absorbing complex was mixed with a more efficient light energy absorbing complex, its absorbance profile would overall increase, making this beneficial for the organism that originally had an inefficient complex.

We agree that red-shifted allophycocyanins (APC-B) would have been nice to include. We do observe a small amount of APC-B in complex with allophycocyanin within one of our

current spectra in Figure 1a. These complexes are < 5% abundance compared with the allophycocyanin peaks observed. To observe whether this APC-B complex can also undergo heterologous complex formation with APC-B variants from other species, we would need to extract and purify, APC-B complexes from multiple species. Although, theoretically this is possible, practically this is a major challenge since conditions would need to be found in all species whereby APC-B variants are highly expressed, and then purification conditions discovered whereby APC-B could be separated readily from its higher abundant allophycocyanin complex. Moreover, due to the high similarity between the APC-B complexes and allophycocyanin, these complexes frequently co-elute using anion exchange chromatography; the dominant chromatographic method for phycocyanin/allophycocyanin purification. Thus, although the result would be interesting, we feel that these studies are not yet feasible and are thus beyond the scope of this manuscript.

Finally, a main take-home message of this manuscript is exemplified by one of their statements: “Thus, this data indicates that if two distinct copies of the phycobiliprotein genes were present and expressed at the same time within a single strain, mixed phycobiliprotein complexes would form.” To me, the mixing of phycobiliproteins *in vitro* resulting in heterogeneous complexes, which has been performed by the authors, does not necessarily mean that would occur *in vivo*. It may, but there may also be some regulatory mechanisms involved in assembly that would make *in vivo* results distinct from *in vitro* results. I suggest that the authors at least acknowledge this possibility and bring it up as discussion. It seems like *in vivo* work would also be a logical next step for the author’s future endeavors.

We fully agree with the reviewer that because we observe this *in vitro*, it does not necessarily mean it would happen *in vivo*. We have expanded the discussion to acknowledge these additional points the reviewer correctly raised.

“Research into whether heterologous complexes can form *in vivo* is a critical next step. Regulatory mechanisms may be involved *in vivo* that either accelerate or prevent heterologous phycobilisome complex assembly.”

Minor:

The authors state “While all known cyanobacteria possess phycobilisomes, the overall structure and size of these complexes can vary widely across different strains.” I suggest adding the word “nearly” before “all”. To my knowledge, there are some *Acaryochloris marina* spp. that lack phycobiliproteins entirely. There are probably others, too.

Thank you for pointing out this mistake that was also pointed out by reviewer 1, we have corrected this now in the main text.

“While most cyanobacteria possess phycobilisomes, the overall structure and size of these complexes can vary widely across different strains.”

The authors state “For instance, rod bundle/paddle-shaped phycobilisomes are found in thylakoid-less cyanobacteria such as *Gloeobacter violaceus*(9, 10), while hemi-discoidal phycobilisomes are the most common(11, 12), with some hemi-ellipsoidal phycobilisomes arrangements also observed.” Since the focus of the paper is on cyanobacteria, I think it is worth mentioning that hemi-ellipsoidal phycobilisomes (and also block-shaped) are only

found in red algae, not cyanobacteria. The authors may want to cite ref. 8 again here which I think covers the most recently known morphologies.

We agree that this will make the introduction clearer. We have now corrected this to:

“For instance, rod bundle/paddle-shaped phycobilisomes are found in thylakoid-less cyanobacteria such as *Gloeobacter violaceus*(9, 10), while hemi-discoidal phycobilisomes are the most common(11, 12), with some hemi-ellipsoidal phycobilisomes arrangements also observed in red algae(8).”

I suggest rearranging the last three paragraphs of the introduction. Halfway through the third to last paragraph, the phrase “here, we...” is used, and then again at the beginning of the last paragraph. Things could be rearranged to have a smoother transition.

We thank the reviewer for pointing this out. We have re-ordered the introduction accordingly.

In section “Phycobiliprotein dynamics vary between phycocyanin rods and allophycocyanin core”, the authors state “Native MS analysis of the phycobiliprotein extract from *L. maxima* shows two dominant charge state distributions between 4500-5500 m/z corresponding to the functional hexameric $(\alpha\beta)_3$ phycocyanin and allophycocyanin complexes (Figure 1a).” I suggest having “hexameric $(\alpha\beta)_3$ ” before both phycocyanin and allophycocyanin.

We have now corrected this to:

“functional hexameric $(\alpha\beta)_3$ phycocyanin and hexameric $(\alpha\beta)_3$ allophycocyanin complexes”

In section “Phycobiliprotein dynamics vary between phycocyanin rods and allophycocyanin core”, the authors state “Despite peaks associated with phycocyanin being more abundant overall...”. Can the authors clarify this? Are they referring to the number of peaks labeled in, for example, Fig. 4A? If so, it looks like the same number of peaks are labeled for PC and APC. There are more PC peaks than APC in Fig. 4C, but there are less PC peaks than APC in, for example, Fig. S4A.

We apologise for this confusion. We are not referring to the number of charge states for the protein but the intensity of the combined PC peaks compared to APC which reflects their overall concentration within the sample. The phycobiliprotein extracts have different concentrations of PC and APC dependent on the composition of their phycobilisome and their phycobilisome architecture at the time of cell lysis and protein extraction. We have reworded the text in this section to make it clearer.

“Despite peaks associated with phycocyanin being overall of higher relative intensity”

In section “Phycobiliprotein dynamics vary between phycocyanin rods and allophycocyanin core”, the authors state “Despite peaks associated with phycocyanin being more abundant overall, the hexameric $(\alpha\beta)_3$ complex is highly dynamic and observed in equilibrium with its dimeric $\alpha\beta$ building block (Figure 1a,b).” I think the authors are specifically referring to Fig. 1B where the blue bars are nearly equal; however, although this is the case for *L. maxima*, it

does not appear to be the case for any of the other species. Maybe my problem is with the author's use of the word equilibrium. Perhaps it would be more appropriate to describe that the ratio of ab dimers being alone or in trimeric complexes is highly variable among species. The APCs being much more commonly in the trimeric form makes sense, so I think that part is fine.

We thank the reviewers for pointing this out that this was not clear. The oligomeric states observed are dependent upon the phycocyanin concentration. We have changed the wording throughout the manuscript to make this clearer. Please refer to the response to the similar comment made by reviewer 1.

To my understanding, pLDDT is not a "measure of the overall uncertainty in the position of each residue" as described by the authors, which makes it seem like B-factor in structural biology. Instead, pLDDT is a confidence metric estimating how well the prediction would agree with an experimental structure. I suggest the authors re-word their description of pLDDT.

We have reworded this accordingly:

"local distance difference test (pLDDT), a measure of the overall uncertainty in the position of each residue a confidence metric estimating how well the prediction is expected to agree with an experimental structure(44, 53)"

Aside from mentioning them in the introduction, the authors do not mention linker proteins. Some APC subunits have the phycocyanobilin-binding domain but also REP domains that serve as linkers. The authors did not test the ability for these to form chimeric complexes with subunits from other species, but this is something that they may want to mention in the discussion as a future direction.

Similarly, the authors may want to mention that assembly/disassembly kinetic experiments could provide insight into the stability and functional relevance of these mixed complexes.

We have now added a small section on linker proteins within the results (see comment to reviewer 1). We have also added the following sentences in the discussion to reflect these future directions.

"Regulatory mechanisms may be involved in vivo that either accelerate or prevent heterologous phycobilisome complex assembly. Moreover, by monitoring the assembly/disassembly kinetics of heterologous complexes with linker proteins present, further insight can be gained into the stability and functional relevance of these complexes in vivo."

Reviewer #3 (Remarks to the Author):

We thank reviewer 3 for assisting with the peer review of this manuscript.

Reviewer #1 (Remarks to the Author):

Overall, the revised version is now suitable for publication, and I thank the authors for the extensive work that went into this revision. I do have one concern, pertaining again to the data for *N. muscorum*. In Fig. S19E-F, the spectroscopic data for this strain show a clear peak at ca. 560 nm, which does not match a "standard" or "typical" PC or APC. Instead, the peak in this region would be either PE or PEC. Does this peak arise from an atypical chromophore composition in PC for this strain, or are there additional phycobiliproteins expressed in this strain under these growth conditions?

Therefore, I recommend publication after minor revision.

We thank the reviewer for this comment. We agree that this strain does not match "typical" PC or APC and likely contains another phycobiliprotein with a different chromophore. These additional phycobiliproteins are expressed, hence the spectroscopic feature we observed; however, they are in low abundance and are not the predominant features in the mass spectra. We have added a note to reflect this in the SI figure legend (Figure S26).

Reviewer #2 (Remarks to the Author):

The authors have sufficiently addressed our comments and we are now happy to recommend this manuscript for publication.

Reviewer #3 (Remarks to the Author):
